# Anti-Cancer Effects of Synergistic Drug–Bacterium Combinations on Induced Breast Cancer in BALB/c Mice

**DOI:** 10.3390/biom9100626

**Published:** 2019-10-18

**Authors:** Menaga Subramaniam, Norhafiza M. Arshad, Kein Seong Mun, Sharan Malagobadan, Khalijah Awang, Noor Hasima Nagoor

**Affiliations:** 1Institute of Biological Sciences (Genetics and Molecular Biology), Faculty of Science, University of Malaya, Kuala Lumpur 50603, Malaysia; menaga_04@yahoo.com (M.S.);; 2Centre for Research in Biotechnology for Agriculture (CEBAR), University of Malaya, Kuala Lumpur 50603, Malaysia; hafizaarshad@um.edu.my; 3Department of Pathology, Faculty of Medicine, University of Malaya, Kuala Lumpur 50603, Malaysia; ksmun@ummc.edu.my; 4Centre for Natural Product Research and Drug Discovery (CENAR) & Department of Chemistry, Faculty of Science, University of Malaya, Kuala Lumpur 50603, Malaysia; khalijah@um.edu.my

**Keywords:** breast cancer, drug combination, nuclear factor κB, inflammatory markers, cytokines

## Abstract

Cancer development and progression are extremely complex due to the alteration of various genes and pathways. In most cases, multiple agents are required to control cancer progression. The purpose of this study is to investigate, using a mouse model, the synergistic interactions of anti-cancer agents, 1′-S-1′-acetoxychavicol acetate (ACA), *Mycobacterium indicus pranii* (MIP), and cisplatin (CDDP) in double and triple combinations to treat chemo-sensitize and immune-sensitize breast cancer. Changes in tumor volume and body weight were monitored. Organs were harvested and stained using hematoxylin–eosin for histopathological assessment. Milliplex enzyme-linked immunosorbent assay (ELISA) was performed to determine cytokine levels, while immunohistochemistry (IHC) was conducted on tumor biopsies to verify systemic drug effects. In vivo mouse models showed tumor regression with maintenance of regular body weight for all the different treatment regimens. IHC results provided conclusive evidence indicating that combination regimens were able to down-regulate nuclear factor kappa-B activation and reduce the expression of its regulated pro-inflammatory proteins. Reduction of pro-inflammatory cytokines (e.g., IL-6, TNF-α, and IFN-ɣ) levels were observed when using the triple combination, which indicated that the synergistic drug combination was able to significantly control cancer progression. In conclusion, ACA, MIP, and CDDP together serve as promising candidates for further development and for subsequent clinical trials against estrogen-sensitive breast cancer.

## 1. Introduction

The mechanisms of cancer development and progression are extremely complex, with the modification of many genes and alteration of various pathways and biochemical activities in the affected cells. Even within a specific cancer type, the presence of heterogeneous cell populations and diverse genetic changes can be detected due to genetic instability [1]. Therefore, a single drug treatment often fails to produce the desired therapeutic effect [2], while combination regimens may provide better cancer treatment.

Screening a novel drug in a cell line system aims to identify the growth rate of cancer cells, the cytotoxicity of the drug, and its mechanism of action [3]. However, a two-dimensional cell culture system alone cannot approximate the complexities of tumor growth and progression in patients. Furthermore, additional genetic and epigenetic changes have also been observed to occur during the propagation of cells in cultures [4,5]. In comparison, tumor cells not only interact with each other but also with the surrounding extracellular matrix and host cells, including endothelial, fibroblasts, immune, and inflammatory cells. It is therefore essential to validate the efficiency of anti-cancer agents in a system which mimics the heterogeneity and interactions of tumor cells. As such, the animal model fits this criteria very well in the investigation of anti-cancer agents.

For the present study, we aimed to investigate the effects of synergistic anti-cancer agent combinations, that is, the chemo-sensitizing properties of 1′-S-1′-acetoxychavicol acetate (ACA), the immune-potentiating activity of heat-killed *Mycobacterium indicus pranii* (MIP), and the cytotoxic effects of cisplatin (CDDP) in an in vivo BALB/c mice model.

The first agent, ACA, is a phenylpropanoid which exhibits anti-cancer effects [6] and has been reported to inhibit the constitutive activation of NF-κB through the suppression of IKKα/β activation [7]. The second agent is the widely used platinum-based anti-cancer drug, CDDP. The third agent, MIP, is a non-pathogenic bacteria known to induce cytotoxicity in various cancer cells [8] and elicit anti-tumor T cell responses [8]. Therefore, we have combined bacterial therapy with chemotherapy to increase the rate of tumor cell death. Felgner and co-workers in 2016 demonstrated that a combination of bacterial therapy and chemotherapy was able to potentiate immune system activation to achieve transient cytotoxic effects while providing long-lasting anti-tumor effects through immunological memory [9]. Heat-killed MIP stimulated cell-mediated responses of the immune system through the induction of CD4^+^ T helper 1 (Th-1) cells in cancer patients [10,11]. The role of MIP in inhibiting NF-κB in highly invasive B16F10 melanoma cells [12] gives valuable insights into the specific pathways involved in this study. Previous in vitro results have shown that the synergistic actions of ACA, MIP, and CDDP in double and triple combinations are able to induce cytotoxicity in MCF-7 breast cancer cells through NF-κB inactivation [9]. Therefore, this study aims to validate these in vitro results in an animal model and observe the effects on the host immune activation.

Double and triple combinations of these agents were administered a in 4T1-induced breast cancer female mouse model. During treatment, the mice body weight, tumor volume, and microscopic changes in major organs were monitored. In addition, immunohistochemistry (IHC) was carried out on tumor sections to identify NF-κB regulated genes and inflammatory biomarkers. The involvement of the immune system was analyzed via quantification of cytokines in blood serum. The results demonstrated that combination treatments have improved anticancer immunotherapy effects compared to the three stand-alone treatments.

## 2. Materials and Methods

### 2.1. Materials

Roswell Park Memorial Institute 1640 (RPMI-1640) media was purchased from Thermo Scientific (Waltham, MA, USA). Fetal bovine serum (FBS) was purchased from Lonza Inc. (Morristown, NJ, USA). Middlebrook 7H10 agar and 7H9 broth were obtained from Sigma-Aldrich (St. Louis, MO, USA). Cisplatin (CDDP) was purchased from Merck (Darmstadt, Germany). 1′S-1′-acetoxychavicol acetate (ACA) was provided by Prof. Dr. Khalijah Awang from the Centre for Natural Product Research and Drug Discovery (CENAR), Department of Chemistry, University of Malaya, Malaysia. *Mycobacterium indicus pranii* was provided by Prof. Dr. Niyaz Ahmed from the Department of Biotechnology and Bioinformatics, School of Life Sciences, University of Hyderabad, India. All other chemicals were of analytical grade and commercially available.

### 2.2. Preparation of 1′-S-1′-Acetoxychavicol Acetate, Mycobacterium indicus pranii, and Cisplatin

ACA and CDDP were prepared fresh in a phosphate-buffered solution (PBS) solution. MIP was cultured in Middlebrook (MB) 7H9 broth supplemented with 0.2% glycerol, 0.05% Tween-80, and 10% albumin-dextrose complex enrichment (ADC). This broth was incubated at 37 °C and 100 rpm agitation until the optical density at 600 nm (OD_600_) reached 1.5. The MIP heat-killed fraction was prepared as described in our previous study [13].

### 2.3. Cell Line and Cell Culture

A highly metastatic breast cancer cell line, 4T1 (Catalogue number: CRL-2539), was purchased from the American Type Culture Collection (ATCC, Manassas, VA, USA). The cell line was cultured at 37 °C and 5% CO_2_ in RPMI-1640 medium supplemented with 10% FBS. The cell line was maintained in low passage to avoid tumor rejection.

### 2.4. Mice Tumor Model and Drug Treatments

Six-week-old female BALB/c mice weighing 15–18 g were fed with sterilized food pellets and water. The mice were purchased from the Animal Ethics Unit of University of Malaya, and the experimental protocols were approved by the Institutional Animal Care and Use Committee (IACUC) of the University of Malaya (Reference number: 2015-181103/IBS/R/MS). All procedures involving laboratory animals were conducted in accordance with the guidelines of the IACUC. There were 7 experimental groups with *n* = 6, as shown in Table 1. For establishing tumors in the mice, 100.0 µl of the 4T1 mouse mammary cells (1 × 10^7^ cells/mL) in 1× PBS were subcutaneously (s.c.) inoculated into the mammary pad region of mice using 25-gauge needles (Becton Dickenson, Franklin Lakes, NJ, USA). Seven days after the inoculation, the mice were randomly divided into seven groups. Once tumor volume reached 100.0 mm^3^, drugs (ACA, MIP, and CDDP) were dissolved in 0.9% (*w*/*v*) NaCl solution and administered subcutaneously at tumor induction sites locally, as shown in Table 1. The mice were treated two times a week at three-days intervals subcutaneously, and blood was collected at the first and fifth week of treatment. Tumor volume was assessed via measurement (length × width × height) with a Traceable Digital Caliper (Thermo Scientific, Waltham, MA, USA) every 7 days from commencement of treatment, and net body weight was measured. After 35 days, the mice were sacrificed, and the tumor and major organs, such as the liver, spleen, kidney, lung, and heart, were harvested for histopathological examination to determine treatment-related microscopic changes. All animal studies were conducted in the Animal Ethics Unit, Faculty of Medicine, University of Malaya. The termination of animals was carried out using purified CO_2_ gas according to the American Veterinary Medical Association (AVMA) guidelines on euthanasia.

### 2.5. Histopathological Examination

The organs were fixed in 10% (*v*/*v*) neutral buffered formalin (NBF) (Merck, Darmstadt, Germany) for 24 h, examined microscopically, and sampled. The sampled sections were processed for paraffin embedding. The embedded tissues were sectioned at a thickness of 5 μm using a microtome, then stained with hematoxylin and eosin (H&E) and evaluated under a light microscope. Histopathological images were captured using an AxioCam MRc5 CCD camera. Organ damage due to toxicity was assessed by Dr. Mun Kein Seong from the Department of Pathology, Faculty of Medicine, University of Malaya. Histopathological evaluations were performed in accordance with the guidelines of the Society of Toxicologic Pathology. To calculate the percentage of necrosis, each tissue/tumor sample was mapped into multiple sections. The sections with complete necrosis were then divided by the total number of sections and multiplied by 100. The number of tumor foci for lung metastases was recorded for each treatment group, and the area of tumor foci was quantified using ImageJ v1.50i software (NIH, Bethesda, MD, USA).

### 2.6. Immunohistochemistry Analysis of Tumor Biopsies

Formalin fixed paraffin embedded (FFPE) tumor sections were subjected to de-paraffinization and rehydration using a graded alcohol series (Cell Signalling, Danvers, MA, USA). Epitope retrieval was achieved by boiling the tissue sections in a 6.0 pH, 0.01M sodium citrate buffer for 10 min, followed by cooling the slides on a bench top for 30 min. Endogenous peroxidase activity was blocked using 3% (*v*/*v*) hydrogen peroxidase (Friedemann Schmidt, Frankfurt, Germany), and the slides were washed with distilled water. All sections were blocked with Tris-buffered saline with Tween-20 (TBST) and 5% (*v*/*v*) normal goat serum (Cell signaling) for 1 h at room temperature. IHC was performed using antibodies specific for NF-κB p65 (1:800), VEGF (1:1600), p21 (1:100), cleaved caspase-3 (1:800), MMP-9 (1:100), and p300 (1:50), and incubation took place overnight at 4 °C. SignalStain^®^ Boost IHC Detection Reagent (HRP-conjugated mouse or rabbit IgG) (Cell Signalling,) was used for signal detection with 3,3′-diaminobenzidine (DAB) solution (Sigma-Aldrich, St. Louis, MO, USA) according to the manufacturer’s protocol. Counter-staining was done using hematoxylin (Sigma-Aldrich), and the slides were thoroughly washed in dH_2_O. The slides were dehydrated through soaking in a graded alcohol series and cleared by soaking in xylene. Slides were then mounted and sealed using dibutyl phthalate xylene (DPX) mounting medium. Images were captured using an inverted microscope Nikon Eclipse TS 100 (Nikon Instruments, Tokyo, Japan), and DAB intensity was quantified using the Nikon NIS-BR Element software (Nikon Instruments, Tokyo, Japan).

### 2.7. Multiplex Assay for the Detection of Cytokines

Blood was obtained from the BALB/c mice using a sterile method in the first and fifth week of treatments. The blood was centrifuged at 1000× *g* and 4 °C for 30 min, and the upper layer containing the serum was collected. Serum samples were prepared for analysis in a 96-well plate using the MILLIPLEX^®^ MAP Mouse Th17 Magnetic Bead Panel kit to detect levels of various cytokines, such as IFN-ɣ, IL-6, IL-2, IL-10, IL-12p70, and TNF-α. Cytokine levels were measured according to the manufacturer’s instructions using a Luminex xMAP system (Luminex Corporation, Austin, TX, USA) with xPONENT software. Standard calibration curves ranged from 7.8 to 8000 pg/mL for IFN-ɣ and IL-6, 6.9–6000 pg/mL for IL-2, 20–20000 pg/mL for IL-10 and IL-12p70, and 3.4–3500 pg/mL for TNF-α.

### 2.8. Statistical Analysis

Data from the experiments were presented as mean ± SD of at least three replicates, and differences between samples were considered statistically significant at *p*-values of <0.01 and <0.05, as calculated using paired Student’s T-test, unless otherwise stated.

## 3. Results

### 3.1. Physiological Effects of MIP, ACA, and CDDP on BALB/c Mice

A subcutaneous injection of 4T1 in the mammary pad of BALB/c was successful as tumor development was noticed from day seven. Figure 1a represents the images of tumors isolated from BALB/c mice after 42 days post-implantation and 35 days post-treatment with various ACA, MIP, and/or CDDP double and triple treatment regimens. The allograft tumors were harvested and photographed. All tested treatment groups resulted in tumor volume regression compared to the placebo, with the triple drug combination treatment being the most effective.

### 3.2. Tumor Volume and Body Weight

Tumor volume and body weight were measured from tumor induction and throughout the treatment period. The results in Figure 1b demonstrated the impact of stand-alone and drug combination treatments on the mean tumor volume and body weight. The mice treated with ACA/MIP/CDDP showed the greatest tumor volume reduction of about 65% compared to the placebo. The second highest tumor volume reduction was observed in the ACA-treated group (52%) followed by the CDDP (46%) and MIP/CDDP (43%) treatment groups. Groups treated with MIP (29%) and ACA/MIP (27%) showed the least reduction. The body weights of mice in all treated groups were maintained at 15–20 g.

### 3.3. Histopathological Evaluation of the Organs

Treatment-related microscopic changes in the major organs were evaluated in hematoxylin and eosin (H&E)-stained paraffin sections, as shown in Figure 2; Figure 3. The 4T1 tumor, being an aggressive malignant cell line, was seen to exhibit widespread metastases in the placebo group. Tumor harvested from MIP-treated groups reached about 50% necrosis, with no obvious squamous or granular differentiation. In CDDP-treated mice, the tumors showed 80% necrosis without obvious squamous differentiation. In the ACA-standalone-treated group, metastases were seen in the heart and lung with 50% necrosis in the tumor mass. In the double combination treatment, metastases in the heart, lung, liver, and spleen were observed with 75% necrosis. In the group treated with the triple combination, metastasis was only seen in the lung with 100% complete necrosis in the tumor nodule. Furthermore, this treatment group showed the lowest number of foci and smallest area of tumor foci in lung metastases compared to the placebo (Table 2 and Figure 4). Therefore, the triple combination treatment group seems to be the most effective in inducing necrosis in the tumor and reducing metastasis.

### 3.4. Immunohistochemistry

In the previous in vitro analysis [13], it was reported that ACA, MIP, and CDDP, as stand-alone treatments and in combination, mediated their anti-cancer effects through NF-κB. IHC was carried out on tumor biopsies to determine the regulation of NF-κB regulated genes and inflammatory biomarkers, such as p65, cleaved caspase-3 (CC-3), matrix metallopeptidase-9 (MMP-9), histone acetyltransferase p300, cyclin-dependent kinase inhibitor p21, and vascular endothelial growth factor (VEGF), on standalone-treated and drug-combination-treated 4T1 tumor biopsies. Figure 5 showed the quantification of the relative intensity of DAB staining in all the IHC analyses on 4T1 tumor sections.

An increase in pro-apoptotic and cell cycle regulator protein levels was observed in p21 and p300. The high expression of p300 was observed in the MIP/ACA-treated group (113.2 ± 5), followed by the ACA (100.3 ± 3.5), MIP/CDDP (99 ± 2.8), and CDDP (95.6 ± 2.6) groups. The expression of p300 in the triple combination group was at a moderate level (94.4 ± 7.4), while the lowest expression was observed in the MIP standalone group (93.7 ± 2.3). The protein expressions of p21 were higher in the ACA treated section (64.9 5 ± 5.23) compared to that treated with the placebo (50.41 ± 1.4). ACA in combination with MIP showed the second highest expression (63.03 ± 3.7), followed by CDDP (56.0 ± 5.9) and MIP/CDDP (56.4 ± 4.8), with the lowest expression seen in the triple combination group (51.6 ± 0.7).

The metastasis biomarker, MMP-9, and angiogenic biomarker, VEGF, levels were reduced compared to those in the placebo. A significant reduction in MMP-9 expression was seen in mice treated with ACA (87.4 ± 3.8) compared to the placebo (137.4 ± 4.9). The second highest reduction was seen in MIP/CDDP-treated mice (102 ± 8.6), followed by mice treated with CDDP (103 ± 10), MIP (109.7 ± 6.8), MIP/ACA (120 ± 0.8), and MIP/ACA/CDDP (132.3 ± 7.6). Therefore, MMP-9 levels were increased in the double and triple drug combination groups but decreased in the ACA standalone group, as was previously seen in another ACA-recombinant human alpha-fetoprotein (rhAFP) combination study [14].

VEGF expression was reduced in all the treatment groups compared to that in the placebo (73.8 ± 2.9). The highest reduction was achieved in the CDDP-treated section (58.21 ± 6.5) followed by the MIP/CDDP (59.6 ± 4.1), MIP (59.7 ± 3.8), and MIP/ACA (62.3 ± 2.7) -treated sections. The smallest reduction was in the ACA-treated tumor section (65.1 ± 5.5). Overall, the drug combination modulated its anti-angiogenic effects consistently both in the in vitro and in vivo analyses.

### 3.5. Cytokine Expression Levels

To determine whether ACA, MIP, and CDDP treatments reduced the release of inflammatory cytokines, such as Th1 cytokines (e.g., IL-2, IL-12, TNF-α, and IFN-ɣ) and Th2 cytokines (e.g., IL-6 and IL-10), cytokine expression levels were analyzed. Cytokine assays revealed distinct cytokine production profiles in each treatment group (Figure 6).

IL-2 was reduced in the placebo and treatment groups on the fifth week compared to the first week, except in ACA-treated mice. The IL-2 expression level was significantly increased from 5.71 ± 1.6 to 10.98 ± 1.3 pg/mL. After treatment with ACA, MIP, CDDP, and MIP/CDDP, IL-12 was increased in the mice blood serum. A significant increase was observed in the groups treated with ACA (10.98 ± 1.3) and CDDP (2.94 ± 0.3) on the fifth week compared to placebo mice (2.6 ± 0.4). Interestingly, its expression during the fifth week was reduced in mice treated with ACA/MIP/CDDP (3.28 ± 0.1) compared to the first week. This may suggest the activation of the immune system in impeding cancer progression through lowering the expression levels of pro-inflammatory markers.

TNF-α expression was comparatively low in all the treatment groups during the first- and fifth-week post-treatment, except in the triple combination group. In this group, TNF-α reached the highest level at 180 ± 4.2 pg/mL during the first week of treatment but was reduced to 11.7 ± 1.2 pg/mL at the end of the treatment course. ACA/MIP-treated mice (7.43 ± 0.3) showed a significant reduction in secreted levels compared to mice in the placebo (6.3 ± 0.2) group during the fifth week.

The IFN-ɣ concentration in the blood serum of ACA/MIP-treated mice (6.42 ± 0.8) was significantly high during the fifth week. However, in the rest of the treated groups, the IFN-ɣ expression level in the blood serum was reduced (3.66 ± 1.2).

In CDDP and triple combination treatment groups, IL-6 was reduced, while increased expression was observed in the other treatment groups. During the first week of CDDP treatment, IL-6 expression reached 80.52 ± 2.5 pg/mL. However, this level was reduced to 44.87 ± 1.8 pg/mL by the fifth week of treatment. Similarly, a 50% reduction in the expression level was found in the ACA/MIP/CDDP group, where the IL-6 level decreased from 104 ± 2.6 to 58 ± 5.5 pg/mL.

In all the treatment groups, the IL-10 expression level was increased compared to that of the placebo. A significant increase of 85.2% was achieved in the ACA treated group, followed by the groups treated with CDDP (60%), ACA/MIP (51%), MIP (47%), MIP/CDDP (45%), and ACA/MIP/CDDP (25%).

## 4. Discussion

In this study, the in vivo mice breast cancer model was carried out to validate the synergistic ability of double and triple anti-cancer agent combinations. In addition, the involvement of NF-κB regulated genes and inflammatory biomarkers, as well as the immune response after treatments, were analyzed. The toxicity effects of ACA as a single drug or in combinations have been reported in earlier research [7,13,14,15]. However, in previous studies, the investigations on ACA and CDDP were in human oral [7], lung, and prostate [14] cancer cell lines in nude mice. Since previous research studies on MIP have been carried out using the BALB/c mice model [8,16,17], the same model was used in this study to investigate the immune-potentiating effect of MIP. However, it required mouse-strain-specific murine tumor cells to induce tumor formation. Therefore, 4T1, a mouse breast tumor cell line, was selected. In addition, the fact that 4T1 breast tumor cells originate from stage IV mice breast cancer and closely mimic human breast cancer is an added advantage for this study. It has been shown that 4T1-induced tumors metastasize spontaneously from the primary tumor in the mammary gland to multiple distant sites, including lymph nodes, blood, liver, lung, brain, and bone [18,19].

Optimized ACA, CDDP [7,14], and MIP [8] standalone treatment doses were obtained from previous in vivo studies, which showed that these doses are safe to be used. For double and combination studies, the quarter maximal inhibitory concentration (IC_25_) values were used since these doses were shown to have synergistic interactions in the in vitro analysis [20].

In our current study, the combination treatments of ACA/MIP, MIP/CDDP, and ACA/MIP/CDDP showed decreased tumor growth in the fifth week compared to the placebo. The body weights of mice taking the ACA/MIP and MIP/CDDP combination treatments were lower after four weeks compared to the placebo group. Regardless, all the treated groups maintained body weights within 15–20 g throughout the treatment period.

Generally, the examination of organs in the placebo mice group showed histopathological changes with respect to the induced tumor. These were taken as the baseline reference. Compared to standalone treatment, tumors in double and triple combination treatment groups showed an increased necrotic effect, with 100% of necrosis observed in the mice who received triple combination treatment.

In the tested model, the MIP standalone treatment did not show a high tumor regression. However, the combination therapy of ACA/MIP/CDDP was able to elicit a significantly stronger response and cause the highest tumor volume reduction. The efficacy of MIP during combination therapy is suggested to be due to its role in targeting the host immune system. As such, the combination of MIP, as an immune potentiator, with ACA and CDDP, as cytotoxic agents, resulted in a synergistic effect against the tumors.

Meanwhile, the combination of ACA and MIP resulted in the least tumor reduction and a higher expression of inflammatory proteins and cytokines, as shown in Figure 5; Figure 6. This suggests that the cytotoxicity induced by CDDP and its regulated signaling pathways are vital for the synergistic effect seen in ACA/MIP/CDDP treatment.

The NF-κB signaling pathway has become a potential target for pharmacological interventions since it regulates the expression of over 500 genes involved in cellular transformation, survival, proliferation, invasion, angiogenesis, metastasis, and inflammation [21,22,23,24]. A number of studies have shown that NF-κB plays a role as a link between inflammation and cancer progression [25,26,27], making NF-κB an essential and potential drug target for hematological malignancies and solid tumors. It was shown in previous studies [7,14] that the expression of NF-κB component p65 and regulators such as VEGF, histone deacetylase 2 (HDAC2), cyclooxygenase-2 (COX-2), 5-lipoxygenase (5-LOX), cyclin-dependent kinase 4 (CDK4), and MMP-9 were reduced in tumors treated with ACA and CDDP. In this study, the focus was on p65, CC-3, MMP-9, p300, p21, and VEGF. When using combination treatments, there was no significant inhibition of p65. This difference in p65 inhibition compared to previous studies using ACA has to be further validated using Western blotting.

Meanwhile, initial metastasis biomarkers, such as VEGF and MMP-9, revealed that combination treatments of MIP, ACA, and CDDP against highly metastatic tumor models successfully downregulated these protein levels. Also, as previously described, ACA, MIP, and CDDP in combination was also shown to enhance apoptosis cell death via intrinsic and extrinsic means and reduce problems related to drug dosage [20]. Additionally, the upregulation of cell cycle-regulated p300 and p21 was also observed. Controlling cell cycle regulation has been shown to be useful in preventing cancer progression [21].

Cytokines are involved in many aspects of cancer, including its development, advancement, treatment, and prognosis, as well as monitoring the effectiveness of cancer treatments. Evidence indicates that cytotoxic anti-cancer agents also affect the immune system, contributing to tumor regression [28,29]. In this study, the expression levels of six important cytokines (i.e., IL-2, IL-6, IL-10, IL-12, TNF-α, and IFN-ɣ) were monitored in six distinct treatment and placebo groups. These cytokines play vital roles in inflammation and cancer progression, including proliferation, migration, and angiogenesis [30,31,32,33,34].

Even though the reduction in the tumor size was similar in the single and double treatment groups, the expressions of inflammatory proteins and cytokines were reduced significantly in the groups which received double combination treatments compared to single treatments, as shown in Figure 5; Figure 6. A reduction in pro-inflammatory cytokines, such as IL-6, TNF-α, and IFN-ɣ, was observed in mice treated with the triple combination, which suggests that a synergistic combination has significant effects against tumor progression. Moreover, the reduction of these cytokine levels may be related to the upregulation of IL-10, since this cytokine can inhibit NF-κB activation and consequently obstruct the production of pro-inflammatory cytokines, including TNF-α, IL-6, and IL-12 [35,36]. The activity of IL-10 was also seen in this study, specifically in the triple combination-treated mice serum, as shown by the decreased levels of IL-6, IL-12, and TNF-α.

## 5. Conclusions

This study indicated that anti-cancer regimens consisting of ACA, MIP, and CDDP in double or triple combinations were successful in enhancing the therapeutic effect by preventing dose-limiting toxicity in breast cancer treatment. It was therefore concluded that the combination of ACA, MIP, and CDDP serves as a promising candidate for further development and subsequent clinical trials for breast cancer.

## Figures and Tables

**Figure 1 biomolecules-09-00626-f001:**
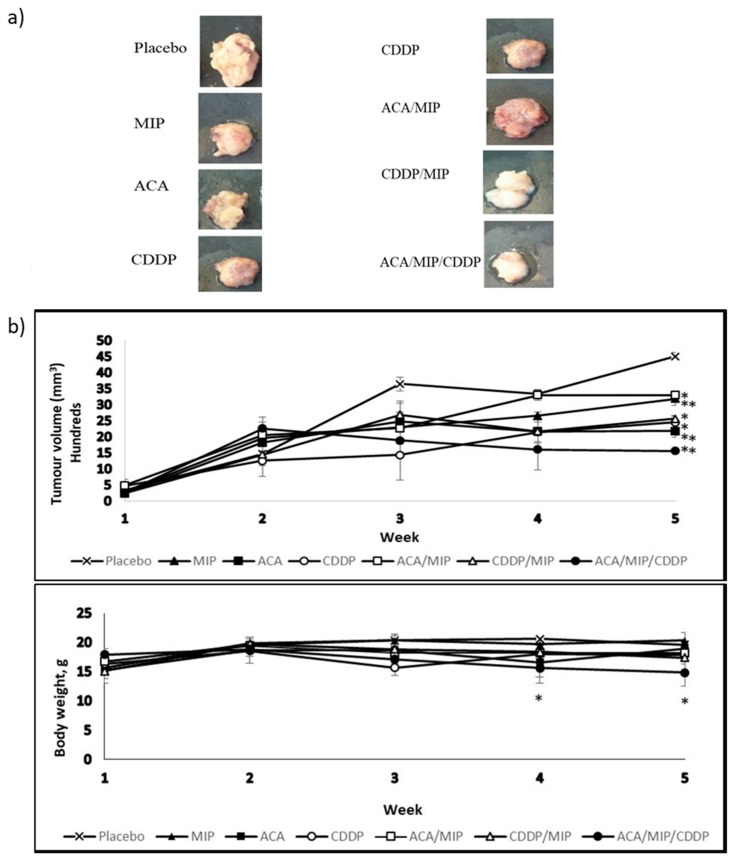
(**a**) Images of tumors isolated from BALB/c mice 35 days post-treatment with a subcutaneous saline injection or with various 1′-S-1′-acetoxychavicol acetate (ACA), *Mycobacterium indicus pranii* (MIP), and cisplatin (CDDP) treatment regimens. All mice were terminated via euthanasia using a flow of pure carbon dioxide (CO_2_) in a gas chamber. Dissections were carried out and tumors were measured and fixed in a 10% (*v*/*v*) neutral buffered formalin (NBF) buffer solution for immunohistochemistry (IHC) analysis. (**b**) Tumor growth and body weight change in 4T1-bearing mice treated with different regimens over a period of five weeks. Tumor volume was calculated from the second week. Data were shown as mean ± standard deviation (SD). Statistically significant changes between the placebo and treatment groups on the fifth week are indicated as * for *p* < 0.05 and ** for *p* < 0.01.

**Figure 2 biomolecules-09-00626-f002:**
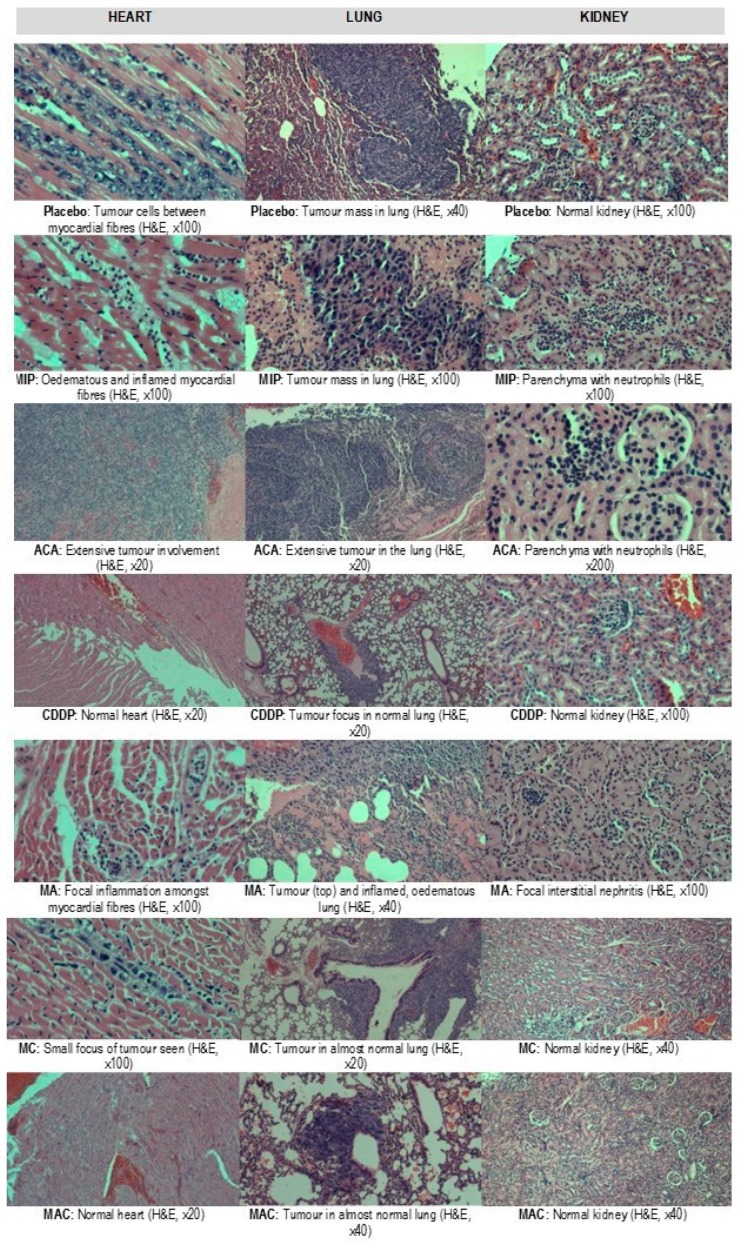
Preliminary toxicity evaluations in the hearts, lungs, and kidneys of BALB/c mice with breast cancer after treatment with placebo, *Mycobacterium indicus pranii* (MIP), 1′-S-1′-acetoxychavicol acetate (ACA), cisplatin (CDDP), ACA/MIP (MA), CDDP/MIP (MC), and ACA/MIP/CDDP (MAC). Hematoxylin and eosin stain. × original magnification.

**Figure 3 biomolecules-09-00626-f003:**
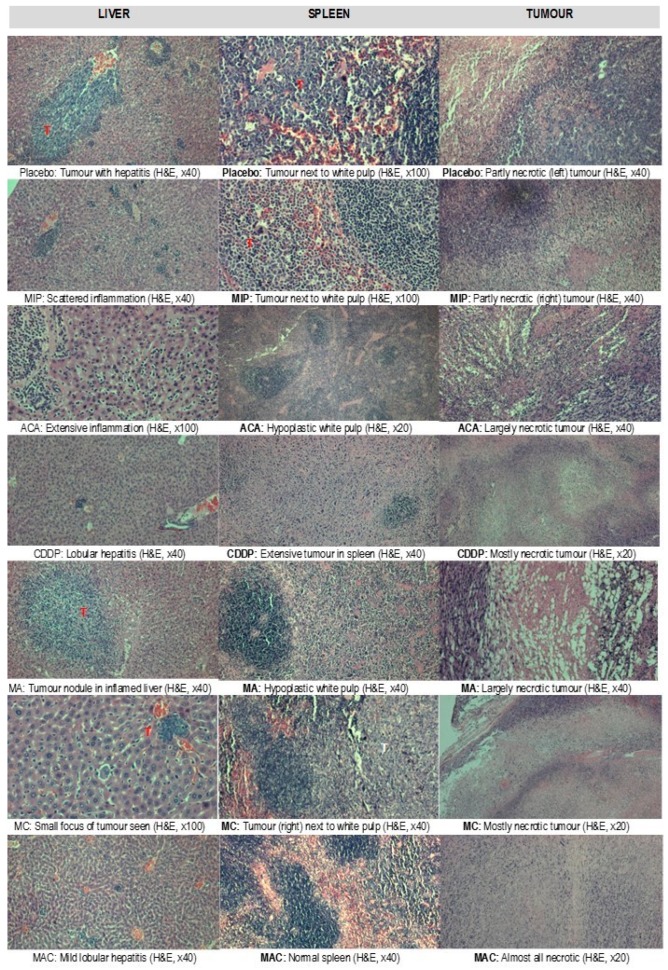
Preliminary toxicity evaluations in the livers, spleens, and tumors (T) of BALB/c mice with breast cancer after treatment with placebo, *Mycobacterium indicus pranii* (MIP), 1′-S-1′-acetoxychavicol acetate (ACA), cisplatin (CDDP), ACA/MIP (MA), CDDP/MIP (MC), and ACA/MIP/CDDP (MAC). Hematoxylin and eosin stain. × original magnification.

**Figure 4 biomolecules-09-00626-f004:**
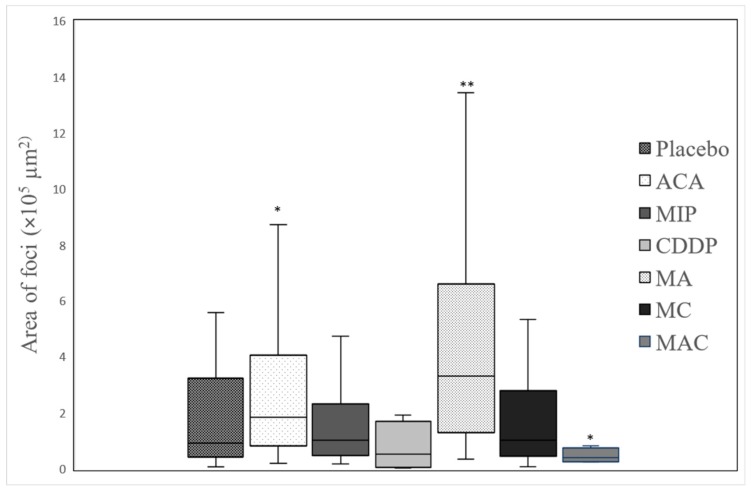
Distribution of the size of tumor foci in lung metastases in BALB/c mice with breast cancer after treatment with a placebo, 1′-S-1′-acetoxychavicol acetate (ACA), *Mycobacterium indicus pranii* (MIP), cisplatin (CDDP), ACA/MIP (MA), CDDP/MIP (MC), and ACA/MIP/CDDP (MAC). Statistically significant differences between placebo and treatment groups after applying the Mann–Whitney test are indicated as * for *p* < 0.05 and ** for *p* < 0.01.

**Figure 5 biomolecules-09-00626-f005:**
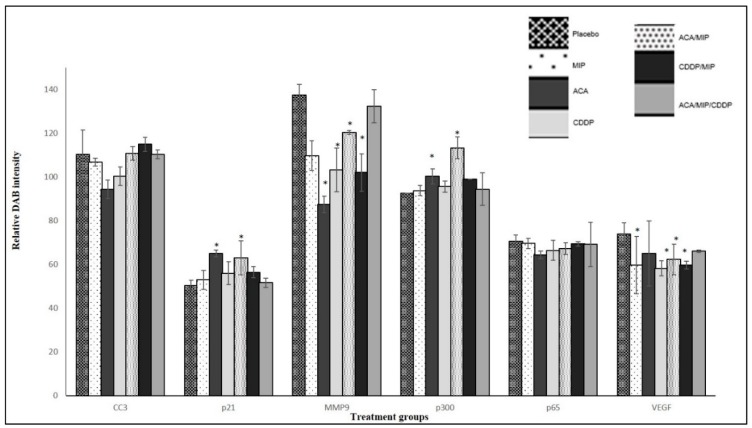
Quantification of the relative intensity of IHC DBA staining on 4T1 breast tumor sections treated with various 1′-S-1′-acetoxychavicol acetate (ACA), *Mycobacterium indicus pranii* (MIP), cisplatin (CDDP) standalone, double, and triple combinations. Images were captured using an inverted Nikon Eclipse TS 100 microscope (Nikon Instruments, Tokyo, Japan), and DAB intensity was quantified using the Nikon NIS-BR Element software (Nikon Instruments, Tokyo, Japan). Data for all NF-κB regulated proteins and inflammatory biomarkers were presented as the mean ± standard deviation (SD) of three independent replicates. Statistically significant changes between placebo and treatment groups at fifth week are indicated as * for *p* < 0.05.

**Figure 6 biomolecules-09-00626-f006:**
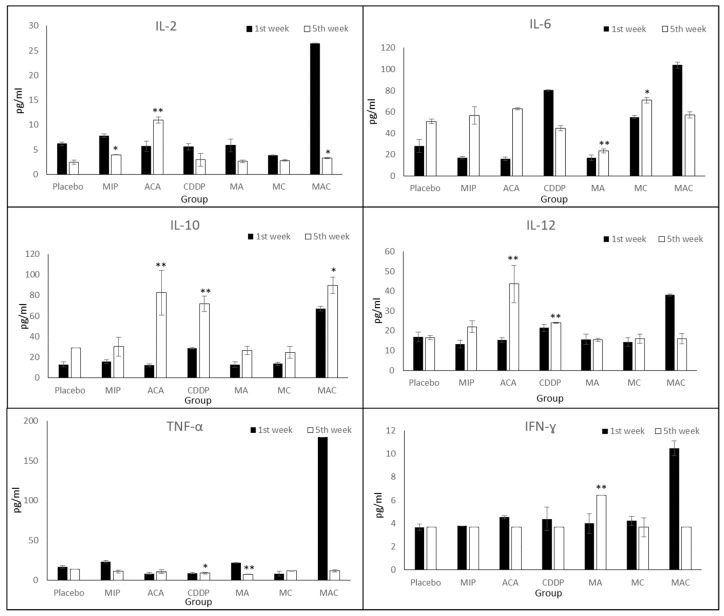
Expression levels of cytokines upon treatment using blood serum on the first and fifth week of the treatment. Treatment groups include placebo, 1′-S-1′-acetoxychavicol acetate (ACA), *Mycobacterium indicus pranii* (MIP), cisplatin (CDDP), ACA/MIP (MA), CDDP/MIP (MC), and ACA/MIP/CDDP (MAC). Statistically significant changes between the placebo and treatment groups on the fifth week are indicated as * for *p* < 0.05 and ** for *p* < 0.01.

**Table 1 biomolecules-09-00626-t001:** The differently treated groups of mice and dosages for the in vivo animal study.

Group (*n* = 6)	Treatment	Treatment Dose
Placebo	Saline	0.9% NaCl
Standalone	MIP	5 × 10^8^ bacilli/mouse [8]
	ACA	1.56 mg/kg [14]
	CDDP	10 mg/kg [14]
Double combination	ACA:MIP	0.4 mg/kg: 1.3 × 10^8^
	CDDP:MIP	2.5 mg/kg: 1.3 × 10^8^
Triple combination	ACA:MIP:CDDP	0.4 mg/kg: 1.3 × 10^8^: 2.5 mg/kg

ACA: 1′-S-1′-acetoxychavicol acetate; MIP: Mycobacterium indicus pranii; CDDP: cisplatin; NaCl: sodium chloride.

**Table 2 biomolecules-09-00626-t002:** Lung metastasis in BALB/c mice with breast cancer after treatment.

Treatment Groups	Number of Foci	Area (Average ± SEM; ×10^5^ µm^2^)
Placebo	52	2.72 ± 0.68
ACA	20	2.94 ± 0.60
MIP	23	1.62 ± 0.30
CDDP	4	0.75 ± 0.39
MA	21	4.13 ± 0.77
MC	45	3.05 ± 0.72
MAC	4	0.46 ± 0.12

Number of foci and average area of lung metastases of BALB/c mice with breast cancer after treatment with a placebo, 1′-S-1′-acetoxychavicol acetate (ACA), *Mycobacterium indicus pranii* (MIP), cisplatin (CDDP), ACA/MIP (MA), CDDP/MIP (MC), and ACA/MIP/CDDP (MAC). Data is presented as mean ± standard error of the mean (SEM).

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
