# Peer review of "Anti-Cancer Effects of Synergistic Drug–Bacterium Combinations on Induced Breast Cancer in BALB/c Mice"

_biomolecules, 2019, doi:10.3390/biom9100626_

Round 1

Reviewer 1 Report

Subramaniam and the co-authors provide evidence for the effects of the combination treatment of ACA, MIP and CDDP in vivo on mouse model representing breast cancer. And they propose that these three anti-cancer agents together serve as promising candidates for further development and subsequent clinical trials for estrogen-sensitive breast cancer. The experiments have been described in sufficient detail and the results are discussed adequately. While generally well done, this reviewer thinks this MS could be accepted after a minor revision. I have included below a number of concerns.

1.     Figure 1 and Figure 4 can be combined because they can show the results better with both the phenotype images and bar data figures together.

2.     The authors should describe why they choose these treatment dose in double and triple combination in Table 1. And why they did not use the ACA and CDDP combination?

3.     Line 178-184: how did the author calculate the percentage of the necrosis as 50% or 80%? Did they do any kind of calculation?

4.     Line 191: what is the “(T)” mean here?

5.     Line 194: use “and” instead of “&”.

6.     Line 201: I do see the body weights of mice were lower in the triple combination group after 4 and 5 weeks than the others’, can the authors explain this?

7.     Line 209: the authors cited wrong reference here. It should be “[9]” rather than “[13]”.

8.     Figure 5: the same question as question 3. What kind of quantification did they used?

9.     Line 216-218: Since there is no significant difference among the control and other treatments, I don’t think the expression of p65 was inhibited.

Reviewer 2 Report

Subramaniam et. al. have assessed the effects of three treatment regimens including- 1’-S-1’acetoxychavicol acetate (ACA), Mycobacterium indicus pranii (MIP) and cisplatin (CDDP) individually or in different combinations to chemosensitize and immunosensitize 4T1 induced breast cancer in-vivo. They measured the tumor volumes, expression of various inflammatory and metastasis markers and serum levels of different cytokines to measure the efficacy of different treatments in reducing breast cancer. They observed regression in tumor volumes, downregulation of NF-κ-B and pro-inflammatory cytokines including IL-6, TNF-α, IFN-γ suggesting a synergistic role of the combination in reducing tumor progression.

Overall, the concept is quite novel in the context of breast cancer. Use of bacterium in treating breast cancer is quite intriguing. However, it raises a lot of questions about the possible side effects resulting from bacterial treatment. There are certain major concerns in the paper that need to be addressed.

1.       This in an in-vivo study that answers a lot of questions about the systemic effects of the tested drugs in regulating the cancer. However, the authors have used only mouse breast cancer cell line 4T1 to address the effects for a drug combination that is intended for possible use in treating human breast cancer. Therefore, including a human breast cancer cell line as orthotopic cancer model and analyzing the effects of the said drugs would add more significance to the study.

2.       Use of MIP in treating breast cancer is quite interesting. However, the results from the article suggest an alternate negative effect of MIP in reducing tumors as most tumors from the group treated with MIP did not have a significant reduction in tumor volume (Page 7 ,line 200).  Furthermore, using ACA in combination with MIP reduced the tumor reduction efficacy of ACA. Likewise, different markers including – p21, MMP9 and p300 had the highest expression in the group treated with ACA/MIP combination. Can the authors provide an explanation for this possible effect of MIP in the context of combination therapy with ACA or by itself?

3.       Figure 4. The results for the tumor volume at the endpoint are contrary to the results shown in figure 1 (tumor images) as tumors from MIP and ACA/MIP are almost similar or slightly bigger in size from the placebo. Can the authors explain the discrepancy in the two results?

4.       The authors have tested the effects of combination of NF-Îş-B mediated signaling without actually assessing the effect on NF-Îş-B itself. It will be better to include expression data from the tumors assessing protein/RNA levels of the transcription factor.

Minor modification:

1.       Page 3, line 106. Please state clearly if the cell line was injected subcutaneously or orthotopically in the mammary fat pad. Please correct the text accordingly in the manuscript.

2.       Figure 1. Please indicate the time points at the tumors shown in the figure were excised and imaged.

3.       Figure 3. Please use same magnification for all the images. If the authors want to show a particular effect, please make an inset with magnified image.

4.       Include the lung metastatic data in the form of a graph to show tumor foci and their sizes to show the effects in different groups in place of using statements as almost normal lung.

5.       Use the abbreviations uniformly across the manuscript. Combination ACA/MIP is stated MA in figure 2 while it written as AM in figure 6. Please correct the text accordingly.

Round 2

Reviewer 2 Report

The authors have included most suggestions in the discussion while making some changes to the figures.